# A New Swiss Federal Act on Gambling: From Missed Opportunities towards a Public Health Approach?

**DOI:** 10.3390/ijerph18126575

**Published:** 2021-06-18

**Authors:** Cheryl Dickson, Emilien Jeannot, Fabio Peduzzi, Jean-Félix Savary, Jean-Michel Costes, Olivier Simon

**Affiliations:** 1Excessive Gambling Center, Addiction Medicine, Lausanne University Hospital (CHUV), Avenue de Morges 10, CH-1004 Lausanne, Switzerland; dr.cheryldickson@live.com (C.D.); emilien.jeannot@chuv.ch (E.J.); 2Faculty of Medicine, Institute of Global Health, Chemin de Mines 9, CH-1202 Geneva, Switzerland; 3Independent Researcher, MPH, CH-1023 Crissier, Switzerland; peduzzi.fabio@gmail.com; 4Groupement Romande d‘Études des Addictions (GREA), CH-1002 Lausanne, Switzerland; jfsavary@grea.ch; 5Observatoire des Jeux, Ministère des Finances, 120 rue de Bercy, CEDEX 12, 75572 Paris, France; jm.costes@orange.fr

**Keywords:** gambling law, public policy, indicators, structural prevention

## Abstract

In January 2019, a new Swiss Federal Act on Gambling (Loi federal de jeux d’argent: LJAr) entered into force following a vote by the Swiss electorate. Intended to modernize and harmonize previous law and open the market for online casinos; the new regulations have highlighted the need for a comprehensive monitoring system. The present article outlines work undertaken by experts within the field to identify and elaborate the first steps towards developing such a monitoring system. This work includes the mapping of institutional actors and draft conceptualization of an impact model, including structural (i.e., prevention and intervention-based components), process (means), and outcomes (effect) indicators. Initial estimations of effective access to indicators and their perceived priority for data gathering are also described. Subsequent steps necessary for implementation of this public health approach for gambling are considered including grey areas for future action.

## 1. Context

The expansion of commercial gambling within recent decades has been met by different regulatory and legislative responses from different countries. Such changes have included the gentle relaxation of regulations in certain jurisdictions and the setting up of government monopolies or regulation and taxation of operators in others [1]. As part of the governments’ responses, attempts to address gambling-related problems through post hoc harm-minimization efforts, known as “responsible gambling” programs, have predominated for many years [2]. Such initiatives place an emphasis on supporting the individual with their gambling-related problems through individualized treatment interventions (self-exclusion, therapy, etc.) [3]. More recently, debate has centralized on the need to introduce a broader public health approach, with a focus on reducing the risks of gambling-related harm to individuals, affected others, and the population at large [4,5]. Aspects of such an approach include increasing an awareness of gambling in order to build public knowledge and resilience, and increasing operator accountability over gambling offers [3]. Alongside this debate, countries, such as Switzerland, have taken the first, tentative steps to adapt and modernize their approach towards gambling; an activity which is widespread across the country, as in most of Europe [6]. Around 46.6% of the Swiss population are believed to gamble each year (73.6% within their lifetime) [7]. This equates to an estimated 5 million people, of whom an estimated 76,000 are reported to gamble excessively [7,8]. The estimated social costs, including healthcare costs, lost productivity, and reduction in quality of life are between 552 and 654 million Swiss Francs (623 to 739 million US Dollars) per year [9]. It is likely that greater numbers are affected by less-severe gambling-related harms, including not only the players themselves but also those close to them, and their communities [10]. In order to promote public health, policies incorporating prevention and harm-reduction features will be necessary to protect the population at large as well as those at risk (so called “prevention paradox”).

In a move towards modernizing and harmonizing pre-existing law and opening the market for online casino gambling, the Swiss electorate voted in the Swiss Federal Act on Gambling (Loi fédérale sur les jeux d’argent; LJAr) in June 2018. The new act, which entered into force in January 2019 [11], recognizes the State’s obligation to adapt structural prevention measures, e.g., operator prevention programs, public education initiatives, etc., in relation to the risk levels of different gambling offers. In order to monitor gambling-related policy, two indicators are currently recognized within the Federal Office of Public Health National Addiction Strategy. These include (1) the number of casino exclusions (data annually produced by the Commission fédérale des maisons de jeu; CFMJ) and (2) excessive gambling prevalence (provided every 5 years by the Swiss Health Survey and mandated by the CFMJ) [12]. Given the many facets of the new law, the need for a more extensive and comprehensive monitoring system is paramount. In the absence of existing literature describing this process, the present article outlines work undertaken as first steps towards developing such a monitoring system. It is hoped that these initial steps can provide a useful overview to those interested in gambling, public health, and monitoring practices. In order to present the current work, a description of the LJAr and its historical context is first given, including its perceived shortfalls. The method and rationale for the work undertaken is then presented. Observations from this work, including the mapping of institutional actors and draft conceptualization of an impact model, is next presented alongside experts’ initial estimates of access to indicators and perceived priorities for data gathering. The final discussion focuses on the subsequent steps necessary for monitoring and potential barriers.

## 2. Evolution of an Approach towards Prevention in the New Law

The LJAr serves the purpose of updating Swiss gambling law for the digital age and improving protection against gambling addiction [13]. Its main aims are, thus, to regulate online gambling, ensure that a portion of gambling income is consecrated for public use, and to ensure that the level of danger associated with different games is taken into account during licensing. The new law extends pre-existing law concerning the obligation for casinos to detect and exclude players who are spending beyond their means (Federal Act on Gambling and Casinos; Loi sur les maisons de jeu; LMJ) [14] and to collaborate with problem-gambling prevention centers (Ordinance on Gambling and Casinos; Ordonnance sur les maisons de jeu; OLMJ) [15]. The 26 separate cantons that make up Switzerland are heavily self-regulated, and additionally to federal law, in 2005 they adopted an intercantonal convention (Intercantonal Convention on the supervision, licensing, and distribution of profits from the lotteries and betting; Convention intercantonale sur les lotteries et le paris; CILP) to impose a 0.5% tax on the revenue from lotteries and sports betting companies for “the prevention and treatment of gambling addiction” (art.18). The cantons also set up a supervisory body for sports betting and lottery operators, known as The Lottery and Betting Board (Comlot), which carries out its duties independently from the existing body for Casino regulation; The Federal Gaming Commission (commission fédérale des maisons de jeu; CFMJ). The new law intends to co-ordinate this dual system for market regulation (CFMJ at a federal level, and Gespa (formerly Comlot) at an intercantonal level). Within its legal framework, the role of both supervisory bodies is detailed, and a third coordinating body, including cantonal and federal representatives, is introduced to resolve discussion points and facilitate collaboration. The new law incorporates regulation of casinos, lotteries (including high-risk electronic lotteries), sports betting and also online gambling, which was previously prohibited. The law does not address microtransactions (discussions over the LJAr began in 2010 and as there are typically delays in bringing such laws into effect, the expanding practices in online gaming and microtransactions are not specifically addressed).

## 3. Social Measures Programs

The LJAr includes a number of measures intended to protect the general population and at-risk individuals against the inherent dangers associated with gambling. Under previous law, casinos were already required to implement a social measures program to protect people who gamble [15]. This included legal obligations to train casino staff on the issue of excessive gambling and to implement exclusion/self-exclusion processes, with imposed fines for noncompliance. In order to strengthen prevention efforts, the current law extends the obligation to implement exclusion processes to online betting operators. In addition, Gespa holds the legal authority to decide whether, and how, such measures will be applied to those offline sports betting games that it deems to be high risk. The effectiveness of such measures must now be reviewed by all gambling operators and included in an annual report submitted to the relevant supervisory body, which also documents the management of conflicts of interest (art. 84). Further requirements to protect players include the stipulation that all games must be operated “safely and transparently”, and games for online use must be specifically designed for accompaniment by protection measures (art. 17). In addition, only authorized Swiss online gambling sites are permitted (unauthorized sites are to be blocked according to art. 86), which should serve to limit gambling opportunities whilst providing legal offers and protecting vulnerable consumers.

As part of the social measures program, casinos, who were already obligated to exclude an individual playing beyond their financial means, are now required to exclude “those people whom they know or should presume, on the basis of an announcement by a specialized service or a social service authority, that they are addicted to gambling” (art. 80). Staff are therefore required to act on their concerns over gambling behavior, regardless of the observed financial outlay. This obligation is also extended to online gambling operators and can also be applied to those offline lottery and sports betting games identified as high risk by Gespa. In addition, casinos and online gambling operators are required to work with a specialist or specialized service in order to lift an exclusion (art. 81). Previous law specified that casinos “must” collaborate with prevention and treatment centers [15]. Whilst current law extends this notion of collaboration to all other games of a large nature, it also weakens this directive stating that gambling operators have “the possibility of” collaborating with actors including prevention and treatment services to develop and implement social measures (art. 76); a point that has caused much concern for intervention services.

## 4. Cantonal Level Interventions

In addition to operator social measures, the cantonal authorities are required to provide prevention and treatment measures for those with a gambling dependency, or people deemed at risk, including people who are close to them (art. 85). As part of the pre-existing framework, the cantons funded prevention programs such as the Intercantonal Program for the Fight Against Gambling Addiction (PILDJ), which is overseen by the Swiss-Romande Group for Addiction Studies (GREA). The 0.5% lottery and sports betting tax, which was initially imposed through cantonal convention (CILP) [16], has been preserved in the LJAr. This source of revenue has largely been used to fund healthcare services, but it is yet to be seen how the prevention tax will now be used to fund prevention and treatment efforts in order to meet this cantonal obligation.

## 5. Other Prevention-Related Provisions

In addition to articles on games licensing, social measures, and prevention by the cantons, the new law and its related ordonnance (OJAr) [17] include specific provisions for the prohibition of advertising targeting minors and/or likely to mislead the public, particularly over the probability of winning (art. 74 LJAr, art. 77 OJAr), limitation of the portion of budgets that operators invest in marketing (art. 22), obligation to submit a conflict of interest management plan (art 81. OJAr), enhanced age control for electronic games operated outside casinos (art 72. LJAr), obligation of operators to submit an annual report on social measures (art. 84 LJAr, art. 86 OJAr), obligations for regulators to make available to researchers the data they have access to in the course of their surveillance activities (art. 76 LJAr, art. 109 OJAr), limiting the remuneration of retailers in relation to turnover, and permitting only an amount that is deemed “reasonable” (art. 46 LJAr). However, the term “reasonable” is left open to interpretation by the new law.

## 6. Limitations of the LJAr

The LJAr implements a new article of the Swiss Constitution stipulating that the State “takes into account the dangers of gambling”. However, the Swiss legislative system involves complex consultation processes, which are typically carried out over several years (nearly 10 years, for the LJAr), and receive heavy lobbying from economic actors, in particular to the operators themselves. The prevention community has seen a significant decline in the scope of the proposed LJAr during its development. The general architecture of the structural prevention measures has been preserved, with three levels of measures. A first group of measures concerns the protection of players by the conditions imposed on operators, largely inspired by the dominant model known as “responsible gambling”. A second group on primary and secondary prevention by the cantons’ health services, which are in charge of public health in Switzerland, was more inspired by the so-called “harm reduction” model anchored in the federal “addictions” policy [12]. A third group concerns the availability of specialized assistance and treatment services, also at the expense of the cantons. Two measures in particular were abolished, weakening the bill: an extra-parliamentary commission to prevent excessive gambling, and additional funding that would have doubled, or even tripled, the resources available in the cantons to implement the second and third groups of measures. The inherent conflict of interest between economy and health has been addressed in theory (the obligation to present a concept addressing this conflict in order to obtain a license), but the conditions of application are unclear and therefore provide little or no practical instruction.

## 7. Method for Identifying Actors and Drafting an Impact Model

To accompany implementation of the new law, initial steps necessary for developing a monitoring system were identified within a workshop held at the Centre for Excessive Gambling (Lausanne University Hospital), in March 2016. This meeting, led by Jean-Michel Costes, Director of l’Observatoire des jeux, France, brought together 20 experts from the fields of law, health and social care, and gambling dependence. In the absence of specific literature-based models, experts drew upon their own theoretical knowledge and professional experiences to identify an approach towards monitoring. The first steps in the monitoring process were identified as (1) mapping the concerned institutions and (2) drafting an impact model, which defines structural, process, and outcome indicators, in order to build the foundations for an effective monitoring system. Each potential indicator was then linked to a group of implicated actors (e.g., operators, regulators, and specialized services) as part of the strategic and operational plan. This information enabled the experts to generate an initial estimation of potential accessibility and perceived priority (high, medium, or low) for each identified indicator.

## 8. Observations

### 8.1. Institutional Actors

Six main types of actors were identified during the mapping work (For a detailed visual representation of this mapping see [18]—Figure 1, page 42) (see Table 1 for an overview). These actors can be incorporated into a comprehensive monitoring system for public policy.

### 8.2. Impact Model

A draft impact model was conceptualized, taking into account previous work on the experiences of other jurisdictions in evaluating gambling prevention [19], the conceptualization of the French impact model [20], and observations by the Working Group on the impact of online gambling in Quebec [21].

The model rests upon three main legal objectives:

That the States “recognize the right of everyone to the enjoyment of the highest attainable standard of physical and mental health” (Article 12 of the United Nation’s international Covenant on Economic, Social and Cultural Rights [22]).

That “the [Swiss] Confederation and the cantons shall take account of the dangers of gambling” (Article 106 of the Federal Constitution of the Swiss Confederation) [23].

That the population should be protected against the dangers inherent to gambling (Article 2; LJAr).

The impact model is derived from the key provisions within the new law, which include primarily operator responsibilities, but also external prevention efforts and healthcare and support. The hypothesized model illustrates how these three main structural resources are associated with specific processes (the proportion of gambling income due to problematic gambling, awareness of the public, individuals, and those close to them, health and social situations of people who gamble problematically) which lead to four key expected outcomes (improvements in quality of life, suicide risk, indebtedness, and social costs). The measurement of each aspect of the system, over time, should therefore provide information about implementation processes and ongoing performance. A diagram of the conceptualized model can be seen in Figure 1.

Experts then drew upon the conceptualized model to identify a range of potential indicators (relating to structures, processes, and outcomes) for policy monitoring. Selection of indicators was based on the criteria that these included (at least in theory) collectible data to evaluate an aspect of the new legal provisions. This led to a list of 32 potential indicators (18 structural, 10 process, and 8 outcome), which can be seen in Table 2). Taken in their entirety, the indicators would provide comprehensive feedback on the LJAr and its facets. The experts’ initial estimations on effective accessibility for the various indicators are categorized as 1. easily accessible, 2. accessible with some difficulties, and 3. extremely difficult to access. The estimated priority for gathering data on each of the indicators is hypothesized to be either; 1. high, 2. medium, or 3. low.

## 9. Discussion and Conclusions

### 9.1. Summary

The draft impact model sets forth the structural resources, associated processes, and expected outcomes to be considered within the proposed monitoring system. The 32 potential indicators derived from this model have been elaborated to enable a comprehensive evaluation of the new law. Work undertaken so far has enabled the identification of an initial approach towards monitoring, the next steps of which will be data collection followed by review. Ratings on the perceived priority and accessibility of indicators will inform the data collection process that will follow (those with high priority and/or relative ease of access being the ideal starting point for evaluation).

### 9.2. Indicator Availability and Accessibility

The subsequent task of gathering and interpreting the data is particularly complex. Some indicators, especially those related to structural resources, can be relatively easily produced and reproduced. These include figures that are simple to generate, such as the capacity for staff training courses, number of staff who attended, and other legally required data, such as operator activity reports. However, other indicators may be more problematic to generate and might require specific research or data-gathering processes to obtain, for example, data on gambling sessions, or figures on the proportion of comorbidities amongst people with problem gambling (as identified by the Swiss Health Survey), compared to samples of those seeking help, and gambling venue clients with no problem gambling. Targeted research projects will be reliant upon funding and dedicated time from the concerned institution(s). The inherent bias of operator-generated research has been well-documented in the literature [25], and thus a global evaluation of this nature can neither be industry-led or based solely on regulator mandates. There is, therefore, a clear need for one or several specialized funds, overseen by peer review, to enable independent and transparent monitoring of policy implementation. As there is currently limited funding available, potential sources need to be clarified.

A second issue is that a lack of collaboration between institutions is likely to limit some aspects of data collection. The mapping process has identified a complex array of institutional actors with differing interests and objectives. Those whose priorities do not fit well with public health-oriented interventions are more likely to be resistant to collaborating, and this has already been seen during certain early attempts to gather data. Differing outlooks, coupled with the weakening of obligations to collaborate, which now permit operators the *choice* to work with treatment and prevention centers (art. 76), are likely to significantly undermine data collection processes. This reflects a key shortfall of the new law, which loads responsibility towards gambling operators. Work will therefore be necessary to build and strengthen relationships, in order to enable collaborative monitoring processes.

In addition, whilst operators are required to submit an annual report on player protection measures (art. 84), they are not legally obliged to collect and submit specific types of data. Without such legal instruction, information on the amounts spent during gambling sessions, for different games, or the nature and motivations for exclusions (imposed or self-exclusion), may not necessarily be collected. This means that the information relating to certain indicators would remain inaccessible.

The current use of two monitoring indicators within the Federal Office of Public Health National Addiction Strategy [12] justifies the need for a more extensive and comprehensive monitoring system. As there is currently no central organization or body providing public health leadership, there is an apparent role for advocacy by professional and interprofessional organizations to lead prevention efforts, raise institutional awareness, and focus priorities. As a precursor to such a structured effort, it will be necessary to secure the interest of actors in the domain and ensure that operator focus shifts from often post hoc “responsible gambling” initiatives to proactive prevention and public safety efforts. Once again, the weakening of legal requirements to collaborate with prevention and treatment services could complicate this effort, leaving operators free to define their own routes and strategies, without independent input from public health professionals. Furthermore, the absence of group representation for people who gamble, including those with a gambling dependency, limits the scope for public input into the development of such prevention initiatives; a point that could be considered in the longer term for empowering consumers, particularly those negatively affected by gambling.

Whilst the LJAr has brought changes to the political landscape, the exact approach towards its implementation is still being defined. It is not yet known, for example, how the cantons will fulfil their obligation to provide a set of prevention and treatment measures, particularly (a) what role existing services will play, (b) how the 0.5% prevention tax on lotteries will be allocated, as part of this requirement, and (c) how publicity restrictions will be addressed (in practice). Similarly, the categorization of electronic games outside of casinos will determine which laws are applicable to these types of games, which are reported to have a high addictive potential [26]. Ongoing debate must determine whether operators should apply similar player protection measures to those used in casinos (e.g., arts. 76, 85).

Finally, strategies to fulfil other legal requirements have not yet been agreed, such as how the safety and transparency of games can be ensured, particularly for online gambling, where legal age restrictions and online self-exclusion will be easy to circumvent. The present article has described the initial steps in developing a monitoring system for the new law. However, an effective system for monitoring must remain sensitive to such developments, and ongoing review and adaptation will be necessary in response to future developments in the field.

## Figures and Tables

**Figure 1 ijerph-18-06575-f001:**
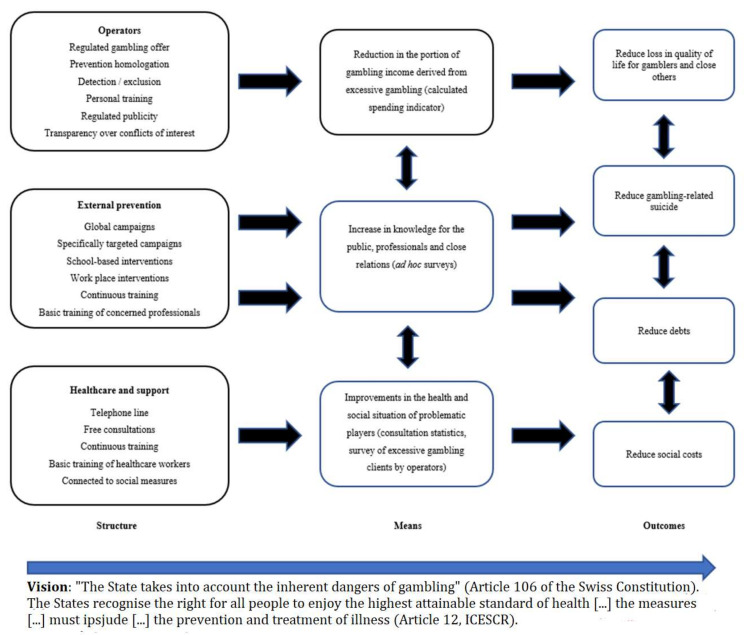
First conceptualization of an impact model.

**Table 1 ijerph-18-06575-t001:** Types of actors.

Actor	Description
Operators	21 land-based casinos; 2 public societies for lottery and sports betting (Loterie Romande and Swisslos); private establishments licensed by the confederation
State surveillance authorities	Commission fédérale des maisons de jeu (CFMJ) and Gespa
State Services	Other federal and cantonal administration services concerned by gambling (economic, financial, legal, and healthcare services)
Prevention and treatment services	Non-governmental organizations and other specialized centers for prevention/treatment
People who gamble	Individuals (currently no group representation)
Beneficiaries	Societies and organizations for sports, social projects, and culture benefitting from gambling revenue

**Table 2 ijerph-18-06575-t002:** A current understanding of identified monitoring indicators.

		Targeted Elements	Indicators	Sources	Effective Accessibility	Priority
**Structural indicators**	**Operators**	Regulated offer	Law/ordinances/regulations; economic data on offers	Regulators	Some difficulties	High
		Standardized prevention	Motivations for regulator decisions/ASTERIG grid or equivalent	Regulators	Extremely difficult	High
		Detection/exclusion	Activity reports	Operators	Some difficulties	Medium
		Training of Personnel	Activity reports	Operator service providers	Some difficulties	Low
		Limiting advertising	Adverts detected as problematic	Media/prevention experts	Extremely difficult	High
		Transparency over conflicts of interest	Mechanisms for personal remuneration	Testimonials	Extremely difficult	High
	**External prevention**	Universal campaigns	Structured concept existing at cantonal/intercantonal level	Intercantonal program	Easy	Low
		Targeted campaigns	Concept existing by canton/intercantonal level with identified groups	Intercantonal program	Some difficulties	Medium
		School interventions	Each cantonal education service has integrated a concept	Cantonal education minister	Some difficulties	High
		Workplace interventions	Ad hoc survey of business panel: N with concept	Professional organizations	Some difficulties	Medium
		Ongoing training	N existing training offers/N people trained	Professional organizations	Some difficulties	High
		Basic training of concerned professions	Explicit objectives in training program catalogs/N dedicated hours	Specialized colleges/faculties	Some difficulties	High
	**Support and healthcare**	Telephone line	N flyers/N posters/N website consultation/N actual calls	Gespa	Easy	Medium
		Free consultations	N places/N consultations	Gespa	Easy	High
		Ongoing training	N existing training offers/N people trained	Professional organizations	Some difficulties	High
		Basic training of socio-health employees	Explicit objectives of training/N dedicated hours	Specialized colleges/faculties	Some difficulties	High
		Adequate remuneration for stakeholders	Average remuneration/related fields—turnover of teams	Specialized services	Extremely difficult	Medium
		Coordination with social measures	Existing ad hoc service contracts under the regulatory authority	Operators/regulators	Some difficulties	High
**Process indicators**		Contribution of gambling dependent players to gambling revenue	(a) Gambling session data	Operators Social support (Enquête suisse sur la santé; ESS)	(a) Extremely difficult	High
			(b) Data from prevalence studies		(b) Easy	Medium
		Public knowledge	Representation survey/5–7 years (possibly online)	Agent/competition	Some difficulties	Medium
		Knowledge of those close to excessive gamblers	Representation survey/5–7 years (possibly online)	Agent/competition	Some difficulties	High
		Knowledge of professionals	Representation survey/5–7 years (possibly online)	Specialized colleges/faculties	Some difficulties	High
		Operator use of social measures	Coverage rate/input–output form	Operators	Some difficulties	Medium
		Use of specialized consultations	Coverage rate/input–output form	Support services	Some difficulties	Medium
		Use of primary care medical services	Proportion of problem gamblers among clients/who broached the subject	General medical personnel	Some difficulties	High
		Use of primary social care services	Proportion of problem gamblers among clients/who broached the subject	Social services	Some difficulties	High
		Health status of people who problem gamble	Proportion of comorbidities among problem gamblers identified in Swiss Health Survey (ESS) versus those seeking support versus gambling venue clients	Social services + healthcare support services + gambling venues	Some difficulties	
**Outcome indicators**		Decreased loss of quality of life for people close to those who problem gamble	Ad hoc survey every 10 years	ESS	Extremely difficult	Medium
		Decreased loss of quality of life for people who problem gamble	Ad hoc survey every 10 years	ESS	Extremely difficult	Medium
		Reduced social costs	Ad hoc survey every 10 years	ESS	Extremely difficult	Medium
		Decrease in gambling-related suicides	Survey of emergencies and specialized units Survey of problem-gambling clients at gaming locations	Healthcare services + Gambling venues	Extremely difficult	High
		Decrease in debt for people who problem gamble	Statistics from healthcare services Debt support service survey Survey of problem-gambling clients at gambling venues	Healthcare services + social services	Some difficulties	High

ASTERIG: Assessment tool to measure and evaluate the risk potential of gambling products [24]. Comlot (Gespa): Intercantonal Commission for Lotteries and Sports Betting. ESS: Swiss Health Survey.

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
