# Peer review of "A New Swiss Federal Act on Gambling: From Missed Opportunities towards a Public Health Approach?"

_ijerph, 2021, doi:10.3390/ijerph18126575_

Round 1

Reviewer 1 Report

Manuscript No.: ijerph-1264551

Manuscript Title: A New Swiss Federal Act on Gambling: Moving towards a Public Health Approach

The manuscript has been improved according to the suggestions of reviewers. The format has been updated and revisions have been made according to the valuable suggestions of the reviewer- all changes and additions highlighted in yellow. But I still suggest some comments for minor revisions.

The page 5, second paragraph, Institutional actors. The text is overly repetitive and exceedingly wordy, there are extensive grammatical issues and a general weakness in the written English.

The conclusion section should be rewritten to better represent a brief overview of general findings of this study.

======================================================

Hope the above comments helpful. Thank you again.

Author Response

Thank you so much for your further feedback which will help to enhance our manuscript. Please find our responses to your comments below.

……………………………………………………………………………………………………………….

  1. The page 5, second paragraph, Institutional actors. The text is overly repetitive and exceedingly wordy, there are extensive grammatical issues and a general weakness in the written English.

In the section entitled “Institutional actors” we aimed to set out the findings from the mapping work. In response to your feedback, we have re-written the section and placed the results within a table in order to present them more succinctly. We have also adjusted the table numbering within the manuscript to take account of this.

  1. The conclusion section should be rewritten to better represent a brief overview of general findings of this study.

In response to your comment we have added a brief summary of the work at the beginning of the “Discussion and conclusion” section. As explained in this section, the work undertaken so far is the start of a long and complex monitoring process, for which the next steps will be data collection and then review. We have therefore retained the main points raised within this section as we believe these are all relevant to the issue of developing a monitoring system.

We hope that we have addressed these matters to your satisfaction.

Thank you again for your much appreciated feedback.

This manuscript is a resubmission of an earlier submission. The following is a list of the peer review reports and author responses from that submission.

Round 1

Reviewer 1 Report

Manuscript No.: ijerph-1218125

Manuscript Title: A New Swiss Federal Act on Gambling: Moving towards a Public Health Approach

Thank you for inviting me to review this manuscript. This paper is an interesting study and has the potential to contribute to the gambling or I would say gaming industry, especially identifying and elaborating the first steps towards developing a monitoring system related to gambling. The current form is a bit hard to follow and read as it tries to put a lot of terms and messages together and in some places have not provided enough explanations especially to the readers from the gaming field. The current form needs significant improvement for further consideration.

My first suggestion is for you to make a more convincing case for the motivation underlying this research.  Based in the literature you cite, I struggled to see how your results extended what we already know – promoting public health, such as policies incorporating prevention and harm-reduction features be necessary, etc. in a country context. What is new in your findings?  In particular, what’s the problem you seek to address? 

Second, the authors put a lot of effort into the diagram of the conceptualized model. However, why did the authors select the external prevention and operators as a starting point? Any theoretical evidence supporting that division especially in the country context? The authors are highly encouraged to completely recheck the information in the description of the conceptualized model, etc.

Third, a total of 32 potential indicators (18 structural, 10 process and 8 outcome) have been derived from the conceptualized impact model. Did you developed these indicators by yourself or were they adapted from previous studies? Authors need to clarify the source of these indicators. The readers may wonder why these 32 selected? What are the criteria?

Fourth, the implications for research and practice are very weak as the results are not very useful for future research. There are no surprising results as the results echoed previous studies.

Fifth, need a further proofreading. Although the paper flows well, there is also a lot of informal language that is used. For example, on page 7, last paragraph. The authors argued that “The monitoring of public policy for gambling has not, as yet, been presented as an activity of particular national interest and the field currently lacks clear public health leadership from a central organization or body.”.

======================================================

Hope the above comments helpful. Thank you again.

Author Response

Thank you for inviting me to review this manuscript. This paper is an interesting study and has the potential to contribute to the gambling or I would say gaming industry, especially identifying, and elaborating the first steps towards developing a monitoring system related to gambling. The current form is a bit hard to follow and read as it tries to put a lot of terms and messages together and in some places have not provided enough explanations especially to the readers from the gaming field. The current form needs significant improvement for further consideration.

Thank you for your feedback, which is extremely helpful. We agree that this article could also be of interest to those in the gaming field. We have taken on board your point about the terminology and have added explanations/sentences to clarify the terminology, throughout the article.

My first suggestion is for you to make a more convincing case for the motivation underlying this research.  Based in the literature you cite, I struggled to see how your results extended what we already know – promoting public health, such as policies incorporating prevention and harm-reduction features be necessary, etc. in a country context. What is new in your findings?  In particular, what’s the problem you seek to address? 

This article aims to inform readers of the provisions within the new Swiss law and work undertaken towards monitoring. We are not aware of literature detailing other work of this nature and therefore hope to provide an insight into the work undertaken. We have added some sentences to the end of the introduction section in order to clarify this point and also touched on the matter in the discussion section.

Second, the authors put a lot of effort into the diagram of the conceptualized model. However, why did the authors select the external prevention and operators as a starting point? Any theoretical evidence supporting that division especially in the country context? The authors are highly encouraged to completely recheck the information in the description of the conceptualized model, etc.

The model is derived from the structure of the new Swiss law (key provisions concern external prevention and operators, and this is reflected in this order which they appear in the model. We therefore prefer to retain the general description of the model but have added a point to clarify the connection between the model and its legal basis. In addition, we have adjusted the wording in this paragraph slightly, which we hope will better explain the model.  

Third, a total of 32 potential indicators (18 structural, 10 process and 8 outcome) have been derived from the conceptualized impact model. Did you developed these indicators by yourself or were they adapted from previous studies? Authors need to clarify the source of these indicators. The readers may wonder why these 32 selected? What are the criteria?

To our knowledge, there were no previous studies into policy monitoring indicators for gambling. The indicators were therefore identified by the 20 experts who took part in a monitoring workshop.  The 32 potential indicators were identified by the group, based on the criteria that they are (at least in theory) collectible data that will enable evaluation of legal provisions. In response to your comment we have added explanations in the sections entitled “Method for identifying and drafting an impact model” and “Indicators*.

In a broader sense, the group also drew upon the experiences of other jurisdictions (working groups) to evaluate gambling prevention and these are referenced in the section entitled “Impact Model”. 

Fourth, the implications for research and practice are very weak as the results are not very useful for future research. There are no surprising results as the results echoed previous studies.

As mentioned in response to point 1, this review paper aims to inform readers of the Swiss context and current work on monitoring, and we are not aware of literature detailing other work of this nature. We have therefore attempted to clarify the purpose and scope of this article by adding some sentences at the end of the introduction section and a further brief point within the discussion section.

Fifth, need a further proofreading. Although the paper flows well, there is also a lot of informal language that is used. For example, on page 7, last paragraph. The authors argued that “The monitoring of public policy for gambling has not, as yet, been presented as an activity of particular national interest and the field currently lacks clear public health leadership from a central organization or body.”.

The paper has now been proofread and we have attempted to formalise the language used. We hope that we have addressed this matter to your satisfaction.

Thank you, once again for your constructive feedback.

Reviewer 2 Report

Thank you for the opportunity to review this interesting article, which provides a clear update of some of the changes to be implemented in Switzerland and sets out some of the challenges of monitoring their impact. This is clearly presented and well set out.

My only query about this article is that it does not critically appraise the approach taken. Whilst the introduction acknowledges a great move towards a public health perspective for gambling, it appears the Swiss reforms are grounded very much in the "responsible gambling" model of prevention. This has been heavily critiqued by public health professionals. I think this article would benefit from including a section about what this legislation misses. From my reading of this, it appears that prevention is to be loaded towards the actions of operators, with universal prevention mainly focused on public health campaigns and education, which are known to have limited efficacy in changing behaviours. Comprehensive universal prevention should include regulation on the types of products provided to an environment, potential limits on how much money could be spent/lost etc, limits on advertising, marketing and sponsorship. Yet, it appears that the new regulations include none of these kinds of provisions. This is an important ommission and I think this article should also critically review the underlying framing of the new legislation as challenge to impact.

Author Response

Thank you for the opportunity to review this interesting article, which provides a clear update of some of the changes to be implemented in Switzerland and sets out some of the challenges of monitoring their impact. This is clearly presented and well set out.

My only query about this article is that it does not critically appraise the approach taken. Whilst the introduction acknowledges a great move towards a public health perspective for gambling, it appears the Swiss reforms are grounded very much in the "responsible gambling" model of prevention. This has been heavily critiqued by public health professionals. I think this article would benefit from including a section about what this legislation misses. From my reading of this, it appears that prevention is to be loaded towards the actions of operators, with universal prevention mainly focused on public health campaigns and education, which are known to have limited efficacy in changing behaviours. Comprehensive universal prevention should include regulation on the types of products provided to an environment, potential limits on how much money could be spent/lost etc, limits on advertising, marketing and sponsorship. Yet, it appears that the new regulations include none of these kinds of provisions. This is an important omission and I think this article should also critically review the underlying framing of the new legislation as challenge to impact.

We fully agree with your perspective and feel that it will greatly enhance the manuscript to include discussion on the issue that you have raised. In response we have added a section entitled “Limitations of the LJAr “. We have also modified the article title and added a brief point on the issue within the “discussion and conclusion” section.